# TraDE: A Simple Self-Attention-Based Density Estimator

## Abstract

We present TraDE, a self-attention-based architecture for auto-regressive density estimation with continuous and discrete valued data. Our model is trained using a penalized maximum likelihood objective, which ensures that samples from the density estimate resemble the training data distribution. The use of self-attention means that the model need not retain conditional sufficient statistics during the auto-regressive process beyond what is needed for each covariate. On standard tabular and image data benchmarks, TraDE produces significantly better density estimates than existing approaches such as normalizing flow estimators and recurrent auto-regressive models. However log-likelihood on held-out data only partially reflects how useful these estimates are in real-world applications. In order to systematically evaluate density estimators, we present a suite of tasks such as regression using generated samples, out-of-distribution detection, and robustness to noise in the training data and demonstrate that TraDE works well in these scenarios.

## 1 Introduction

Density estimation involves estimating a probability density $p(x)$, given independent, identically distributed (iid) samples from it. This is a versatile and important problem as it allows one to generate synthetic data or perform novelty and outlier detection. It is also an important subroutine in applications of graphical models. Deep neural networks are a powerful function class and learning complex distributions with them is promising. This has resulted in a resurgence of interest in the classical problem of density estimation.

One of the more popular techniques for density estimation is to sample data from a simple reference distribution and then to learn a (sequence of) invertible transformations that allow us to adapt it to a target distribution. Flow-based methods (Durkan et al., 2019b) employ this with great success. A more classical approach is to decompose $p(x)$ in an iterative manner via conditional probabilities $p(x_{i+1}|x_{1...i})$ and fit this

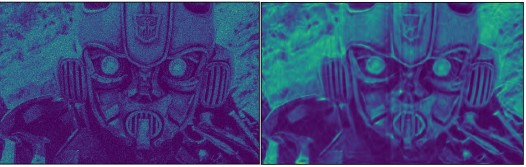

Figure 1: TraDE is well suited to density estimation of Transformers. Left: Bumblebee (true density), Right: density estimated from data.

distribution using the data (Murphy, 2013). One may even employ implicit generative models to sample from $p(x)$ directly, perhaps without the ability to compute density estimates. This is the case with Generative Adversarial Networks (GANs) that reign supreme for image synthesis via sampling (Goodfellow et al., 2014; Karras et al., 2017).

Implementing these above methods however requires special care, e.g., the normalizing transform requires the network to be invertible with an efficiently computable Jacobian. Auto-regressive models using recurrent networks are difficult to scale to high-dimensional data due to the need to store a potentially high-dimensional conditional sufficient statistic (and also due to vanishing gradients). Generative models can be difficult to train and GANs lack a closed density model. Much of the current work is devoted to mitigating these issues. The main contributions of this paper include:

1. We introduce TraDE, a simple but novel auto-regressive density estimator that uses self-attention along with a recurrent neural network (RNN)-based input embedding to approximate arbitrary continuous and discrete conditional densities. It is more flexible than contemporary architectures such as those proposed by Durkan et al. (2019b;a); Kingma et al. (2016); De Cao et al. (2019) for

this problem, yet it remains capable of approximating *any* density function. To our knowledge, this is the first adaptation of Transformer-like architectures for continuous-valued density estimation.

2. Log-likelihood on held-out data is the prevalent metric to evaluate density estimators. However, this only provides a partial view of their performance in real-world applications. We propose a suite of experiments to systematically evaluate the performance of density estimators in downstream tasks such as classification and regression using generated samples, detection of out-of-distribution samples, and robustness to noise in the training data.

3. We provide extensive empirical evidence that TraDE substantially outperforms other density estimators on standard and additional benchmarks, along with thorough ablation experiments to dissect the empirical gains. The main feature of this work is the simplicity of our proposed method along with its strong (systematically evaluated) empirical performance.

## 2 BACKGROUND AND RELATED WORK

Given a dataset $\{x^1, \ldots, x^n\}$ where each sample $x^l \in \mathbb{R}^d$ is drawn iid from a distribution $p(x)$, the maximum-likelihood formulation of density estimation finds a $\theta$-parameterized distribution $q$ with

$$\widehat{\theta} = \underset{\theta}{\operatorname{argmax}} \frac{1}{n} \sum_{l=1}^{n} \log q(x^l; \theta). \tag{1}$$

The candidate distribution $q$ can be parameterized in a variety of ways as we discuss next.

**Normalizing flows** write $x \sim q$ as a transformation of samples $z$ from some base distribution $p_z$ from which one can draw samples easily (Papamakarios et al., 2019). If this mapping is $f_\theta : z \to x$, two distributions can be related using the determinant of the Jacobian as

$$q(x; \theta) := p_z(z) \left| \frac{\mathrm{d}f_\theta}{\mathrm{d}z} \right|^{-1}.$$

A practical limitation of flow-based models is that $f_\theta$ must be a diffeomorphism, i.e., it is invertible and both $f_\theta$ and $f_\theta^{-1}$ are differentiable. Good performance using normalizing flows imposes nontrivial restrictions on how one can parametrize $f_\theta$: it must be flexible yet invertible with a Jacobian that can be computed efficiently. There are a number of techniques to achieve this, e.g., linear mappings, planar/radial flows (Rezende & Mohamed, 2015; Tabak & Turner, 2013), Sylvester flows (Berg et al., 2018), coupling (Dinh et al., 2014) and auto-regressive models (Larochelle & Murray, 2011). One may also compose the transformations, e.g., using monotonic mappings $f_\theta$ in each layer (Huang et al., 2018; De Cao et al., 2019).

**Auto-regressive models** factorize the joint distribution as a product of univariate conditional distributions $q(x; \theta) := \prod_i q_i(x_i | x_1, \ldots x_{i-1}; \theta)$. The auto-regressive approach to density estimation is straightforward and flexible as there is no restriction on how each conditional distribution is modeled. Often, a single recurrent neural network (RNN) is used to sequentially estimate all conditionals with a shared set of parameters (Oliva et al., 2018; Kingma et al., 2016). For high-dimensional data, the challenge lies in handling the increasingly large state space $x_1, \ldots, x_{i-1}$ required to properly infer $x_i$. In recurrent auto-regressive models, these conditioned-upon variables' values are stored in some representation $h_i$ which is updated via a function $h_{i+1} = g(h_i, x_i)$. This overcomes the problem of high-dimensional estimation, albeit at the expense of loss in fidelity. Techniques like masking the computational paths in a feed-forward network are popular to alleviate these problems further (Uria et al., 2016; Germain et al., 2015; Papamakarios et al., 2017).

Many existing auto-regressive algorithms are highly sensitive to the variable ordering chosen for factorizing $q$, and some methods must train complex ensembles over multiple orderings to achieve good performance (Germain et al., 2015; Uria et al., 2014). While autoregressive models are commonly applied to natural language and time series data, this setting only involves variables that are already naturally ordered (Chelba et al., 2013). In contrast, we consider continuous (and discrete) density estimation of vector valued data, e.g. tabular data, where the underlying ordering and dependencies between variables is often unknown.

**Generative models** focus on drawing samples from the estimated distribution that look resemble the true distribution of data. There is a rich history of learning explicit models from variational inference (Jordan et al., 1999) that allow both drawing samples and estimating the log-likelihood or implicit models such as Generative Adversarial Networks (GANs, see (Goodfellow et al., 2014))

where one may only draw samples. These have been shown to work well for natural images (Kingma & Welling, 2013) but have not obtained similar performance for tabular data. We also note the existence of many classical techniques that are less popular in deep learning, such as kernel density estimation (Silverman, 2018) and Chow-Liu trees (Chow & Liu, 1968; Choi et al., 2011).

## 3 TOOLS OF THE TRADE

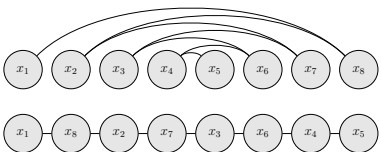

Consider the 8-dimensional Markov Random Field shown here, where the underlying graphical model is unknown in practice. Consider the following two orders in which to factorize the autoregressive model: $(1, 2, 3, 4, 5, 6, 7, 8)$ and $(1, 8, 2, 7, 3, 6, 4, 5)$. In the latter case the model becomes a simple sequence where e.g. $p(x_3|x_{1,8,2,7}) = p(x_3|x_7)$ due to conditional independence. A latent variable auto-regressive model only needs to preserve the most recently encountered state in this latter ordering. In the first ordering, $p(x_3|x_{1,2})$ can be simplified further to $p(x_3|x_2)$, but we still need to carry the precise value of $x_1$ along until the end since $p(x_8|x_{1...7}) = p(x_8|x_{1,2})$. This is a fundamental weakness in models employing RNNs such as (Oliva et al., 2018). In practice, we may be unable to select a favorable ordering for columns in a table (unlike for language where words are inherently ordered), especially as the underlying distribution is unknown.

### 3.1 VERTEX ORDERING AND SUFFICIENT STATISTICS

The above problem is also seen in sequence modeling, and Transformers were introduced to better model such long-range dependencies through self-attention (Vaswani et al., 2017). A recurrent network can, in principle, absorb this information into its hidden state. In fact, Long-Short-Term Memory (LSTM) (Hochreiter & Schmidhuber, 1997) units were engineered specifically to store long-range dependencies until needed. Nonetheless, storing information costs parameter space. For an auto-regressive factorization where the true conditionals require one to store many variables' values for many time steps, the RNN/LSTM hidden state must be undesirably large to achieve low approximation error, which means these models must have many parameters (Collins et al., 2017). The following simple lemma formalizes this.

**Lemma 1.** *Denote by $G$ the graph of an undirected graphical model over random variables $x_1, \ldots, x_d$. Depending on the order vertices are traversed in our factorization the largest number of latent variables a recurrent auto-regressive model needs to store is bounded from above and below by the minimum and the maximum number of variables with a cut edge of the graph $G$.*

**Proof**. Given a subset of known variables $S \subseteq \{1, \ldots d\}$ we want to estimate the conditional distribution of the variables on the complement $C := \{1, \ldots d\} \backslash S$. For this we need to decompose $S$ into the Markov blanket $M$ of $C$ and its remainder. By definition $M$ consists of the variables with a cut edge. Since $p(x_C|x_S) = p(x_C|x_M)$ we are done. ∎

This problem with long-dependencies in auto-regressive models has been noted before, and is exacerbated in density estimation with continuous-valued data, where values of conditioned-upon variables may need to be precisely retrieved. For instance, recent auto-regressive models employ masking to eliminate the sequential operations of recurrent models (Papamakarios et al., 2017). There are also models like Pixel RNN (Oord et al., 2016) which explicitly design a multi-scale masking mechanism suited for natural images (and also require discretization of continuous-valued data). Note that while there is a natural ordering of random variables in text/image data, variables in tabular data do not follow any canonical ordering.

Applicable to both continuous and discrete valued data, our proposed TraDE model circumvents this issue by utilizing self-attention to retrieve feature values relevant for the conditioning, which does *not* require more parameters to retrieve the values of relevant features that happened to appear early in the auto-regressive factorization order (Yun et al., 2019). Thus a major benefit of self-attention here is its effectiveness at maintaining an accurate representation of the feature values $x_j$ for $j < i$ when inferring $x_i$ (irrespective of the distance between $i$ and $j$).

## 3.2 The Architecture of TraDE

TraDE is an auto-regressive density estimator and factorizes the distribution $q$ as

$$q(x;\theta) := \prod_{i=1}^{d} q_i(x_i|x_1,\ldots,x_{i-1};\theta); \qquad (2)$$

Here the $i^{\text{th}}$ univariate conditional $q_i$ conditions the feature $x_i$ upon the features preceding it, and may be easier to model than a high-dimensional joint distribution. Parameters $\theta$ are shared amongst conditionals. Our main observation is that auto-regressive conditionals can be accurately modeled using the attention mechanism in a Transformer architecture (Vaswani et al., 2017).

**Self-attention.** The Transformer is a neural sequence transduction model and consists of a multi-layer encoder/decoder pair. We only need the encoder for building TraDE. The encoder takes the input sequence $(x_1,\ldots,x_d)$ and predicts the $i^{\text{th}}$-conditional $q_i(x_i|x_1,\ldots,x_{i-1};\theta)$. For discrete data, each conditional is parameterized as a categorical distribution. For continuous data, each conditional distribution is parametrized as a mixture of Gaussians, where the mean/variance/proportion of each mixture component depend on $x_1,\ldots,x_{i-1}$:

$$q_i(x_i|x_1,\ldots,x_{i-1};\theta) = \sum_{k=1}^{m} \pi_{k,i}\, N(x_i;\mu_{k,i},\sigma_{k,i}^2). \qquad (3)$$

with the mixture proportions $\sum_{k=1}^{m} \pi_{k,i} = 1$. All three of $\pi_{k,i}, \mu_{k,i}$ and $\sigma_{k,i}$ are predicted by the model as separate outputs at each feature $i$. They are parametrized by $\theta$ and depend on $x_1,\ldots,x_{i-1}$.

The crucial property of the Transformer's encoder is the self-attention module outputs a representation that captures correlations in different parts of its input. In a nutshell, the self-attention map outputs $z_i = \sum_{j=1}^{d} \alpha_j \varphi(x_j)$ where $\varphi(x_j)$ is an embedding of the feature $x_j$ and normalized weights $\alpha_j = \langle \varphi(x_i, \varphi(x_j) \rangle$ compute the similarity between the embedding of $x_i$ and that of $x_j$. Self-attention therefore amounts to a linear combination of the embedding of each feature with features that are more similar to $x_i$ getting a larger weight. We also use multi-headed attention like (Vaswani et al., 2017) which computes self-attention independently for different embeddings. Self-attention is the crucial property that allows TraDE to handle long-range and complex correlations in the input features; effectively this eliminates the vanishing gradient problem in RNNs by allowing direct connections between far away input neurons (Vaswani et al., 2017). Self-attention also enables permutation equivariance and naturally enables TraDE to be agnostic to the ordering of the features.

*Masking* is used to prevent $x_i, x_{i+1}, \ldots, x_d$ from taking part in the computation of the output $q_i$ and thereby preserve the auto-regressive property of our density estimator. We keep residual connections, layer normalization and dropout in the encoder unchanged from the original architecture of (Vaswani et al., 2017). The final output layer of our TraDE model, is a position-wise fully-connected layer where, for continuous data, the $i^{\text{th}}$ position outputs the mixing proportions $\pi_{k,i}$, means $\mu_{k,i}$ and standard deviations $\sigma_{k,i}$ that together specify a Gaussian mixture model. For discrete data, this final layer instead outputs the parameters of a categorical distribution.

**Positional encoding** involves encoding the position $k$ of feature $x_k$ as an additional input, and is required to ensure that Transformers can identify which values correspond to which inputs – information that is otherwise lost in self-attention. To circumvent this issue, Vaswani et al. (2017) append Fourier position features to each input. Picking hyper-parameters for the frequencies is however difficult and it does not work well either for density estimation (see Sec. 4). An alternative that we propose is to use a simple recurrent network at the input to embed the input values at each position. Here the time-steps of the RNN implicitly encode the positional information, and we use a Gated Recurrent Unit (GRU) model to better handle long-range dependencies (Cho et al., 2014). This parallels recent findings from language modeling where (Wang et al., 2019) also used an initial RNN embedding to generate inputs to the transformer. Observe that the GRU does not slow down TraDE at inference time since sampling is performed in auto-regressive fashion and remains $\mathcal{O}(d)$. Complexity of training is marginally higher but this is more than made up for by the superior performance.

**Remark 2 (Compared to other methods, TraDE is simple and effective).** Our architecture for TraDE can be summarized simply as using the encoder of a Transformer (Vaswani et al., 2017) with appropriate masking to achieve auto-regressive dependencies with an output layer consisting of a mixture of multi-variate Gaussians and an input embedding layer built using an RNN. And

yet it is more flexible than architectures for normalizing flows without restrictive constraints on the input-output map (e.g. invertible functions, Jacobian computational costs, etc.) As compared to other auto-regressive models, TraDE can handle long-range dependencies and does not need to permute input features during training/inference like Germain et al. (2015); Uria et al. (2014). Finally, the objective/architecture of TraDE is general enough to handle both continuous and discrete data, unlike many existing density estimators. As experiments in Sec. 4 shows these properties make TraDE very well-suited for auto-regressive density estimation.

Unlike discrete language data, tabular datasets contain many numerical values and their categorical features do not share a common vocabulary. Thus existing applications of Transformers to such data remain limited. More generally, successful applications of Transformer models to continuous-valued data as considered here have remained rare to date (without coarse discretization as done for images in Parmar et al. (2018)), limited to a few applications in speech (Ren et al., 2019) and time-series (Li et al., 2019; Benidis et al., 2020; Wu et al., 2020).

Beyond Transformers' insensitivity to feature order when conditioning on variables, they are a good fit for tabular data because their lower layers model lower-order feature interactions (starting with pairwise interactions in the first layer and building up in each additional layer). This relates them to ANOVA models (Wahba et al., 1995) which only progressively blend features unlike in the fully-connected layers of feedforward neural networks.

**Lemma 3.** *TraDE can approximate any continuous density $p(x)$ on a compact domain $x \in \mathcal{X} \subset \mathbb{R}^d$.*

**Proof.** Consider any auto-regressive factorization of $p(x)$. Here the $i^{\text{th}}$ univariate distribution $p(x_i \mid x_1, \ldots, x_{i-1})$ can be approximated arbitrarily well by a Gaussian mixture $\sum_{k=1}^{m} \pi_{k,i} N(x_i; \mu_{k,i}, \sigma_{k,i}^2)$, assuming a sufficient number of mixture components $m$ (Sorenson & Alspach, 1971). By the universal approximation capabilities of RNNs (Siegelmann & Sontag, 1995), the GRU input layer of TraDE can generate nearly any desired positional encodings that are input to the initial self-attention layer. Finally, Yun et al. (2019) establish that a Transformer architecture with positional-encodings, self-attention, and position-wise fully connected layers can approximate any continuous sequence-to-sequence function over a compact domain. Their constructive proof of this result (Theorem 3 in Yun et al. (2019)) guarantees that, even with our use of masking to auto-regressively restrict information flow, the remaining pieces of the TraDE model can approximate the mapping $\{(x_1, \ldots, x_{i-1})\}_{i=1}^{d} \rightarrow \{(\pi_{k,i}, \mu_{k,i}, \sigma_{k,i}^2)\}_{i=1}^{d}$. ∎

### 3.3 THE LOSS FUNCTION OF TRADE

The MLE objective in (1) does not have a regularization term to deal with limited data. Furthermore, MLE-training only drives our network to consider how it is modeling each conditional distribution in (2), rather than how it models the full joint distribution $p(x)$ and the quality of samples drawn from its joint distribution estimate $q(x; \widehat{\theta})$. To encourage both, we combine the MLE objective with a regularization penalty based on the Maximum Mean Discrepancy (MMD) to get the loss function of TraDE. The former ensures consistency of the estimate while the MMD term is effective in detecting obvious discrepancies when the samples drawn from the model do not resemble samples in the training dataset (details in Appendix A). The objective minimized in TraDE is a penalized likelihood:

$$L(\theta) = -\frac{1}{nd} \sum_{l=1}^{n} \sum_{i=1}^{d} \log q_i(x_i^l | x_1^l, \ldots, x_{i-1}^l; \theta) + \lambda \, \text{MMD}^2(\widehat{p}(x), q(x; \theta)) \quad (4)$$

where hyper-parameter $\lambda \geq 0$ controls the degree of regularization and $\widehat{p}$ denotes the empirical data distribution. This objective is minimized using stochastic gradient methods, where the gradient of the log-likelihood term can be computed using standard back-propagation. Computing the gradient of the MMD term involves differentiating the samples from $q_\theta$ with respect to the parameters $\theta$. For continuous-valued data, this is easily done using the reparametrization trick (Kingma & Welling, 2013) since we model each conditional as a mixture of Gaussians. For categorical features, we calculate the gradient using the Gumbel softmax trick (Maddison et al., 2016). The TraDE is thus general enough to handle both continuous and discrete data, unlike many existing density estimators.

We also show experiments in Appendix B where MMD regularization helps density estimation with limited data (beyond using other standard regularizers like weight decay and dropout); this is common in biological applications, where experiments are often too costly to be extensively replicated (Krishnaswamy et al., 2014; Chen et al., 2020). Finally we note that there exist a large

number of classical techniques, such as maximum entropy and approximate moment matching techniques (Phillips et al., 2004; Altun & Smola, 2006) for regularization in density estimation; MMD is conceptually close to moment matching.

## 4 EXPERIMENTS

We first evaluate TraDE both qualitatively (Fig. 2 and Fig. S1 in the Appendix) and quantitatively on standard benchmark datasets (Sec. 4.1). We then present three additional ways to evaluate the performance of density estimators in downstream tasks (Sec. 4.3) along with some ablation studies (Sec. 4.2 and Appendix A.2). Details for all the experiments in this section, including hyper-parameters, are provided in the Appendix D.

**Remark 4 (Reporting log-likelihood for density estimators).**
The objective in (1) is a maximum likelihood objective and therefore the maximizer need not be close to $p(x)$. This is often ignored in the current literature and algorithms are compared based on their log-likelihood on held-out test data. However, the log-likelihood may be insufficient to ascertain the real-world performance of these models, e.g., in terms of verisimilitude of the data they generate. This is a major motivation for us to develop a complementary evaluation methodologies in Sec. 4.3. We emphasize **these evaluation methods are another novel contribution of our paper**. However, we also report log-likelihood results for comparison with others.

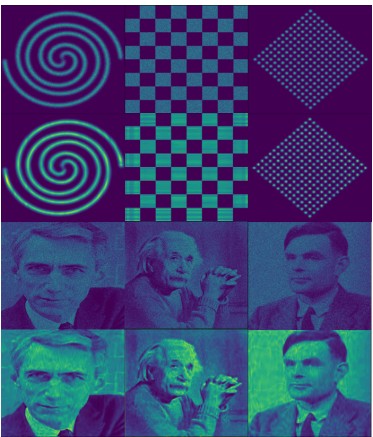

Figure 2: **Qualitative evaluation on 2-dimensional datasets.** We train TraDE on samples from six 2-dimensional densities and evaluate the model likelihood over the entire domain by sampling a fine grid; original densities are shown in rows 1 & 3 and estimated densities are shown in rows 2 & 4. This setup is similar to (Nash & Durkan, 2019). These distributions are highly multi-modal with complex correlations but TraDE learns an accurate estimate of the true density across a large number of examples.

### 4.1 RESULTS ON BENCHMARK DATASETS

We follow the experimental setup of (Papamakarios et al., 2017) to ensure the same training/validation/test dataset splits in our evaluation. In particular, preprocessing of all datasets is kept the same as that of (Papamakarios et al., 2017). The datasets named POWER, GAS (Vergara et al., 2012), HEPMASS, MINI-BOONE and BSDS300 (Martin et al., 2001) were taken from the UCI machine learning repository (Dua & Graff, 2017).

Table 1: **Negative average test log-likelihood in nats (smaller is better) on binarized MNIST.**

|  | LOG-LIKELIHOOD |
|---|---|
| VAE | $82.14 \pm 0.07$ |
| PLANAR FLOWS | $81.91 \pm 0.22$ |
| IAF | $80.79 \pm 0.12$ |
| SYLVESTER | $80.22 \pm 0.03$ |
| BLOCK NAF | $80.71 \pm 0.09$ |
| PIXELRNN | $79.20$ |
| TRADE (OURS) | $\mathbf{78.92 \pm 0.00}$ |

The MNIST dataset (LeCun et al., 1990) is used to evaluate TraDE on high-dimensional image-based data. We follow the variational inference literature, e.g., (Oord et al., 2016), and use the binarized version of MNIST. The datasets for anomaly detection tasks, namely Pendigits, ForestCover and Satimage-2 are from the Outlier Detection DataSets (OODS) library (Rayana, 2016). We normalized the OODS data by subtracting the per-feature mean and dividing by the standard deviation. We show the results on benchmark datasets in Table 2. There is a wide diversity in the algorithms for density estimation but we make an effort to provide a complete comparison of known results irrespective of the specific methodology. Some methods like Neural Spline Flows (NSF) by (Durkan et al., 2019b) are quite complex to implement; others like Masked Autoregressive Flows (MAF) (Papamakarios et al., 2017) use ensembles to estimate the density; some others like Autoregressive Energy Machines (AEM) of (Nash & Durkan, 2019) average the log-likelihood over a large number of importance samples. As the table shows, TraDE obtains performance improvements over all these methods in terms of the log-likelihood. This performance is persistent across all datasets except MINIBOONE where TraDE is competitive although not the best. The improvement is drastic for the POWER, HEPMASS and BSDS300.

We also evaluate TraDE on the MNIST dataset in terms of the log-likelihood on test data. As Table 1 shows TraDE obtains high log-likelihood even compared to sophisticated models such as Pixel-RNN (Oord et al., 2016), VAE (Kingma & Welling, 2013), Planar Flows (Rezende & Mohamed,

Table 2: **Average test log-likelihood in nats (higher is better) for benchmark datasets.** Entries marked with [*] evaluate standard deviation across 3 independent runs of the algorithm; all others are mean $\pm$ standard error across samples in the test dataset. TraDE achieves significantly better log-likelihood than other algorithms on all datasets except MINIBOONE.

| | POWER | GAS | HEPMASS | MINIBOONE | BSDS300 |
|---|---|---|---|---|---|
| REAL NVP (DINH ET AL., 2016) | $0.17 \pm 0.01$ | $8.33 \pm 0.14$ | $-18.71 \pm 0.02$ | $-13.84 \pm 0.52$ | $153.28 \pm 1.78$ |
| MADE MOG (GERMAIN ET AL., 2015) | $0.4 \pm 0.01$ | $8.47 \pm 0.02$ | $-15.15 \pm 0.02$ | $-12.27 \pm 0.47$ | $153.71 \pm 0.28$ |
| MAF MOG (PAPAMAKARIOS ET AL., 2017) | $0.3 \pm 0.01$ | $9.59 \pm 0.02$ | $-17.39 \pm 0.02$ | $-11.68 \pm 0.44$ | $156.36 \pm 0.28$ |
| FFJORD (GRATHWOHL ET AL., 2018) | $0.46 \pm 0.00$ | $8.59 \pm 0.00$ | $-14.92 \pm 0.00$ | $-10.43 \pm 0.00$ | $157.4 \pm 0.00$ |
| NAF[*] (HUANG ET AL., 2018) | $0.62 \pm 0.01$ | $11.96 \pm 0.33$ | $-15.09 \pm 0.4$ | $\mathbf{-8.86 \pm 0.15}$ | $157.43 \pm 0.3$ |
| TAN (OLIVA ET AL., 2018) | $0.6 \pm 0.01$ | $12.06 \pm 0.02$ | $-13.78 \pm 0.02$ | $-11.01 \pm 0.48$ | $159.8 \pm 0.07$ |
| BNAF[*] (DE CAO ET AL., 2019) | $0.61 \pm 0.01$ | $12.06 \pm 0.09$ | $-14.71 \pm 0.38$ | $-8.95 \pm 0.07$ | $157.36 \pm 0.03$ |
| NSF[*] (DURKAN ET AL., 2019B) | $0.66 \pm 0.01$ | $13.09 \pm 0.02$ | $-14.01 \pm 0.03$ | $-9.22 \pm 0.48$ | $157.31 \pm 0.28$ |
| AEM (NASH & DURKAN, 2019) | $0.70 \pm 0.01$ | $13.03 \pm 0.01$ | $-12.85 \pm 0.01$ | $-10.17 \pm 0.26$ | $158.71 \pm 0.14$ |
| TRADE [*] (OURS) | $\mathbf{0.73 \pm 0.00}$ | $\mathbf{13.27 \pm 0.01}$ | $\mathbf{-12.01 \pm 0.03}$ | $-9.49 \pm 0.13$ | $\mathbf{160.01 \pm 0.02}$ |

2015), IAF (Kingma et al., 2016), Sylvester (Berg et al., 2018), and Block NAF (De Cao et al., 2019). This is a difficult dataset for density estimation because of its high dimensionality. We also show the quality of the samples generated by our model in Fig. S1.

## 4.2 ABLATION STUDY

We begin with an RNN trained as an auto-regressive density estimator; the performance of this basic model in Table 3 is quite poor for all datasets. Next, we use a Transformer network trained in auto-regressive fashion without position encoding; this

Table 3: **Average test log-likelihood in nats (higher is better) on benchmark datasets for ablation experiments.**

| | POWER | GAS | HEPMASS | MINIBOONE | BSDS300 |
|---|---|---|---|---|---|
| RNN | 0.51 | 6.26 | -15.87 | -13.13 | 157.29 |
| Transformer (without position encoding) | 0.71 | 12.95 | -15.80 | -22.29 | 134.71 |
| Transformer (with position encoding) | 0.73 | 12.87 | -13.89 | -12.28 | 147.94 |
| TraDE without MMD | 0.72 | 13.26 | -12.22 | -9.44 | 159.97 |

leads to significant improvements compared to the RNN but not on all datasets. Compare this to the third row which shows the results for Transformer with position encoding (instead of the GRU-based embedding in TraDE); this performs much better than a Transformer architecture without position encoding. This suggests that incorporating the information about the position is critical for auto-regressive models (also see Sec. 3). A large performance gain on all datasets is observed upon using a GRU-based embedding that replaces the position embedding of the Transformer.

Note that although the MMD regularization helps produce a sizeable improvement for HEPMASS, this effect is not consistent across all datasets. This is akin to any other regularizer; regularizers do not generally improve performance on all datasets and models. MMD-penalization of the MLE objective additionally helps obtain higher-fidelity samples that are similar to those in the training dataset; this helps when few data are available for density estimation which we study in Appendix B. This is also evident in the regression experiment (Table 4) where the MSE of TraDE is significantly better than MLE-based RNNs and Transformers trained without the MMD term.

## 4.3 SYSTEMATIC EVALUATION OF DENSITY ESTIMATORS

We propose evaluating density estimation in four canonical ways, regression using the generated samples, a two-sample test to check the quality of generated samples, out-of-distribution detection, and robustness of the density estimator to noise in the training data. This section shows that TraDE performs well on these tasks which demonstrates that it not only obtains high log-likelihood on held-out data but can also be readily used for downstream tasks.

**1. Regression using generated samples.** We first characterize the quality of samples generated by the model, based on a regression task where $x_d$ is regressed using data from the others $(x_1, \ldots, x_{d-1})$.

The procedure is as follows: first we use the training set of the HEPMASS ($d = 21$) to fit the density estimator, and then create a synthetic dataset with both inputs $z = (x_1, \ldots, x_{d-1})$ and targets $y = x_d$ sampled from the model. Two random forest regressors are fitted, one on the real data and another

on this synthetic data. These regressors are tested on real test data from HEPMASS. If the model synthesizes good samples, we expect that the test performance of the regressor fitted on synthetic data would be comparable to that of the regressor fitted on real data. Table 4 shows the results. Observe that the classifier trained on data synthesized by TraDE performs very similarly to the one trained on the original data. The MSE of a RNN-based auto-regressive density estimator, which is higher, is provided for comparison. The MSE of a standard Transformer with position embedding is much worse at 1.37.

**2. Two-sample test on the generated data.** A standard way to ascertain the quality of the generated samples is to perform a two-sample test (Wasserman, 2006). The idea is similar to a two-sample test (Lopez-Paz & Oquab, 2016) in the discriminator of a GAN: if the samples generated by the auto-regressive model are good, the discriminator should have an accuracy of 50%. We train a density estimator on the training data; then fit a random forest classifier to differentiate between real validation data and synthesized validation data; then compute the accuracy of the classifier on the real test data and the synthesized test data. The accuracy using TraDE is $51 \pm 1\%$ while the accuracy using RNN is $55 \pm 4\%$ and that of a standard Transformer with position embedding is much worse at $88 \pm 15\%$. Standard deviation is computed across different subsets of features $x_1, (x_1, x_2), \ldots, (x_1, \ldots, x_d)$ as inputs to the classifier. This suggests that samples generated by TraDE are much closer those in the real dataset than those generated by the RNN model.

Table 4: **Mean squared error of regression on HEPMASS.**

| | |
|---|---|
| Real data | 0.773 |
| TraDE | 0.780 |
| RNN | 0.803 |
| Transformer | 1.37 |

**3. Out-of-distribution detection.** This is a classical application of density estimation techniques where we seek to discover anomalous samples in a given dataset. We follow the setup of (Oliva et al., 2018): we call a datum out-of-distribution if the likelihood of a datum $x$ under the model $q(x; \theta) \leq t$ for a chosen threshold $t \geq 0$. We can compute the average precision of detecting out-of-distribution samples by sweeping across different values of $t$. The results are shown in Table 5. Observe that TraDE obtains extremely good performance, of more than 0.95 average precision, on the three datasets.

Table 5: **Average precision for out-of-distribution detection** The results for NADE, NICE and TAN were (carefully) eye-balled from the plots of (Oliva et al., 2018).

| | NADE | NICE | TAN | TraDE |
|---|---|---|---|---|
| Pendigits | 0.91 | 0.92 | 0.97 | 0.98 |
| ForestCover | 0.87 | 0.80 | 0.94 | 0.95 |
| Satimage-2 | 0.98 | 0.975 | 0.98 | 1.0 |

**4. TraDE builds robustness to noisy data.** Real data may contain noise and in order to be useful on downstream tasks. Density estimation must be insensitive to such noise but the maximum-likelihood objective is sensitive to noise in the training data. Methods such as NSF (Durkan et al., 2019b) or MAF (Papamakarios et al., 2017) indirectly mitigate this sensitivity using permutations of the input data or masking within hidden layers but these operations are not designed to be robust to noisy data. We study how TraDE deals with this scenario. We add noise to 10% of the entries in the training data; we then fit both TraDE and NSF on this noisy data; both models are evaluated on clean test data. As Table 6 shows, the degradation of both TraDE and NSF is about the same; the former obtains a higher log-likelihood as noted in Table 2.

Table 6: **Average test log-likelihood (nats) for HEP-MASS dataset with and without additive noise in the training data.**

| | Clean Data | Noisy Data |
|---|---|---|
| NSF | -14.51 | -14.98 |
| TraDE | -11.98 | -12.43 |

## 5 DISCUSSION

This paper demonstrates that self-attention is naturally suited to building auto-regressive models with strong performance in continuous and discrete valued density estimation tasks. Our proposed method is a universal density estimator that is simpler and more flexible (without restrictions of invertibility and tractable Jacobian computation) than architectures for normalizing flows, and it also handles long-range dependencies better than other auto-regressive models based on recurrent structures. We contribute a suite of downstream tasks such as regression, out of distribution detection, and robustness to noisy data, which evaluate how useful the density estimates are in real-world applications. TraDE demonstrates state-of-the-art empirical results that are better than many competing approaches across several extensive benchmarks.

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

# Appendix

# TraDE: A Simple Self-Attention-Based Density Estimator

## A   REGULARIZATION USING MAXIMUM MEAN DISCREPANCY (MMD)

An alternative to maximum likelihood estimation, and in some cases a dual to it, is to perform non-parametric moment matching (Altun & Smola, 2006). One can combine a log-likelihood loss and a two-sample discrepancy loss to ensure high fidelity, i.e., the samples resemble those from the original dataset.

We can test whether two distributions $p$ and $q$ supported on a space $\mathcal{X}$ are different using samples drawn from each of them by finding a smooth function that is large on samples drawn from $p$ and small on samples drawn from $q$. If $x \sim p$ and $y \sim q$, then $p = q$ if and only if $\mathbb{E}_x \left[ f(x) \right] = \mathbb{E}_y \left[ f(y) \right]$ for all bounded continuous functions $f$ on $\mathcal{X}$ (Lemma 9.3.2 in (Dudley, 2018)). We can exploit this result computationally by restricting the test functions to some class $f \in \mathcal{F}$ and finding the worst test function. This leads to the Maximum Mean Discrepancy metric defined next (Fortet & Mourier, 1953; Müller, 1997; Gretton et al., 2012; Sriperumbudur et al., 2016). For a class $\mathcal{F}$ of functions $f : \mathcal{X} \to \mathbb{R}$, the MMD between distributions $p, q$ is

$$\text{MMD}[\mathcal{F}, p, q] = \sup_{f \in \mathcal{F}} \left( \mathbb{E}_{x \sim p} \left[ f(x) \right] - \mathbb{E}_{y \sim q} \left[ f(y) \right] \right). \tag{5}$$

It is cumbersome to find the supremum over a general class of functions $\mathcal{F}$ to compute the MMD. We can however restrict $\mathcal{F}$ to be the unit ball in a universal Reproducing Kernel Hilbert Space (RKHS) (Gretton et al., 2012)) with kernel $k$. The MMD is a metric in this case and is given by

$$\text{MMD}^2[k, p, q] = \mathbb{E}_{x,x' \sim p} \left[ k(x, x') \right] - 2 \mathbb{E}_{x \sim p, y \sim q} \left[ k(x, y) \right] + \mathbb{E}_{y,y' \sim p} \left[ k(y, y') \right] \tag{6}$$

With a universal kernel (e.g. Gaussian, Laplace), MMD will capture *any* difference between distributions  (Steinwart, 2001). We can easily obtain an empirical estimate of the MMD above using samples (Gretton et al., 2012).

### A.1   GRADIENT OF MMD TERM

Evaluating the gradient of the MMD term (6) involves differentiating the samples from $q_\theta$ with respect to the parameters $\theta$. When TraDE handles continuous variables, it samples from mixture of Gaussians for each conditional hence gradient of the MMD term is done using the reparametrization trick (Kingma et al., 2015). On the other hand, gradient of the MMD term (6) is computed using the Gumbel softmax trick (Maddison et al., 2016; Jang et al., 2016) for discrete variables (i.e., binarized MNIST). The objective of TraDE is thus general enough to easily handle both continuous and discrete data distributions which is not usually the case with other methods.

### A.2   WHAT IF ONLY MMD IS USED AS A LOSS FUNCTION?

In theory, MMD with a universal kernel would also produce *consistent* estimates (Gretton et al., 2009), but maximizing likelihood ensures our estimate is statistically *efficient* (Daniels, 1961; Loh & Wainwright, 2013). Although a two-sample discrepancy loss like MMD can help produce samples that resemble those from the original dataset, using MMD as the only objective function will not result in a good density estimator given limited data. To test this hypothesis, we run experiments in which only MMD is utilized as a loss function without the maximum-likelihood term.

Note that using our MMD term with a fixed kernel (without solving a min-max problem to optimize the kernel) only implies that the distributions match up to the chosen kernel. In other words, the global minima of our MMD objective with a fixed kernel are not necessarily the global minima of the maximum likelihood term, an additional reason for the poor NLLs as shown in Table S1. While the MLE and MMD objectives should both result in a consistent estimator of $p(x)$ in the limit of infinite data and model-capacity, in finite settings these objectives favor estimators with different properties. In practice a combination of both objectives yields superior results, and makes performance less dependent on the MMD kernel bandwidth.

Table S1: Test likelihood (higher is better) using **ONLY** the MMD objective for training vs. TraDE.

| Dataset | Only MMD | TraDE |
|---|---|---|
| POWER | -5.00 ± 0.24 | **0.73 ± 0.00** |
| GAS | -10.85 ± 0.16 | **13.27 ± 0.01** |
| HEPMASS | -28.18 ± 0.12 | **-12.01 ± 0.03** |
| MINIBOONE | -68.06 | **-9.49 ± 0.13** |
| BSDS300 | 70.70 ± 0.32 | **160.01 ± 0.02** |

## B  DENSITY ESTIMATION WITH FEW DATA

MMD regularization especially helps when we have few training data. Test likelihoods (higher is better) after training on a sub-sampled MINIBOONE dataset are: -55.62 vs. -55.53 (100 samples) and -49.07 vs. -48.90 (500 samples) for TraDE without and with MMD-penalization, respectively. Density estimation from few samples is common in biological applications, where experiments are often too costly to be extensively replicated (Krishnaswamy et al., 2014; Chen et al., 2020).

## C  SAMPLES FROM TRADE TRAINED ON MNIST

We also evaluate TraDE on the MNIST dataset in terms of the log-likelihood on test data. As Table 1 shows TraDE obtains high log-likelihood even compared to sophisticated models such as Pixel-RNN (Oord et al., 2016). This is a difficult dataset for density estimation because of the high dimensionality. Fig. S1 shows the quality of the samples generated by TraDE.

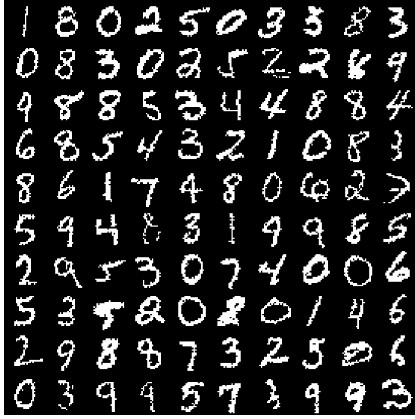

Figure S1: **Samples from TraDE fitted on binary MNIST.**

## D  HYPER-PARAMETERS

All models are trained for 1000 epochs with the Adam optimizer (Kingma & Ba, 2014). The MMD kernel is a mixture of 5 Gaussians for all datasets, i.e. $k(x, y) = \sum_{j=1}^{5} k_j(x, y)$ where each $k_j(x, y) = e^{-\|x-y\|_2^2/\sigma_j^2}$ with bandwidths $\sigma_j \in \{1, 2, 4, 8, 16\}$ (Li et al., 2017). Each layer of TraDE consists of a GRU followed by self-attention block which consists of multi-head-self-attention and fully connected feed-forward layer. A layer normalization and a residual connection is used around each of these components (i.e. multi-head-self-attention and fully connected) in self-attention block (Vaswani et al., 2017). Multiple layers of self-attention block can be stacked together in TraDE. Finally, the output layer of TraDE outputs $\pi_{k,i}, \mu_{k,i}$, and $\sigma_{k,i}$ which specify a univariate Gaussian mixture model approximation of each conditional distribution in the auto-regressive factorization. Table S2 shows TraDE hyper-parameters. We used a coarse random search based on validation NLL to select suitable hyper-parameters. Note that dataset used in this work have very different number of samples and dimensions as shown in Table S3

Table S2: **Hyper-parameters for benchmark datasets.**

|  | POWER | GAS | HEPMASS | MINIBOONE | BSDS300 | MNIST |
|---|---|---|---|---|---|---|
| MMD coefficient $\lambda$ | 0.2 | 0.1 | 0.1 | 0.4 | 0.2 | 0.1 |
| Gaussian mixture components | 150 | 100 | 100 | 20 | 100 | 1 |
| Number of layers | 5 | 8 | 6 | 8 | 5 | 6 |
| Multi-head attention head | 8 | 16 | 8 | 8 | 2 | 4 |
| Hidden neurons | 512 | 400 | 128 | 64 | 128 | 256 |
| Dropout | 0.1 | 0.1 | 0.1 | 0.2 | 0.3 | 0.1 |
| Learning rate | 3E-4 | 3E-4 | 5E-4 | 5E-4 | 5E-4 | 5E-4 |
| Mini-batch size | 512 | 512 | 512 | 64 | 512 | 16 |
| Weight decay | 1E-6 | 1E-6 | 1E-6 | 0 | 1E-6 | 1E-6 |
| Gradient clipping norm | 5 | 5 | 5 | 5 | 5 | 5 |
| Gumbel softmax temperature | n/a | n/a | n/a | n/a | n/a | 1.5 |

Table S3: **Dataset information**. In order to ensure that results are comparable and same setups and data splits are used as previous works, we closely follow the experimental setup of (Papamakarios et al., 2017) for training/validation/test dataset splits. In particular, the preprocessing of all the datasets is kept the same as that of (Papamakarios et al., 2017). Note that, these setups were used on all other baselines as well.

| Dataset | dimension | Training size | Validation Size | Test size |
|---|---|---|---|---|
| POWER | 6 | 1659917 | 184435 | 204928 |
| GAS | 8 | 852174 | 94685 | 105206 |
| HEPMASS | 21 | 315123 | 35013 | 174987 |
| MINIBOONE | 43 | 29556 | 3284 | 3648 |
| BSDS300 | 63 | 1000000 | 50000 | 250000 |
| MNIST | 784 | 50000 | 10000 | 10000 |
| ForestCover | 10 | 252499 | 28055 | 5494 |
| Pendigits | 16 | 5903 | 655 | 312 |
| Satimage-2 | 36 | 5095 | 566 | 142 |

