# OpenReview forum: "TraDE: A Simple Self-Attention-Based Density Estimator"
_ICLR.cc/2021/Conference — Reject_

### Official Review · AnonReviewer2 · 2020-10-27

**Rating:** 3
**Confidence:** 5

**Review:**

This paper proposes TraDE, a transformer-based density estimator that is capable of learning a density of real-valued tabular data. Compared to previously proposed transformers, there are three main differences in TraDE model: 1) the output is modeled as a mixture of Gaussians, 2) maximum mean discrepancy (MMD) is added to the loss, and 3) Gated Recurrent Unit (GRU) is used to provide positional encoding. Tested on a suite of benchmark tasks, the proposed method shows promising results over baselines.

Strengths:

The paper is the first to apply transformers on continuous-valued tabular data. The motivation of using transformers for auto-regressive modeling of continuous data stated in Section 3 is persuasive.

In addition to reporting the test likelihood, the authors conduct supplementary experiments to validate the effectiveness of the proposed model. The experiments include training a regression model with generated samples, two-sample testing using a classifier, detecting out-of-distribution samples, and learning on noise-corrupted data. These analyses provide a rich view of how the proposed model behaves and confirms the effectiveness of the proposed approach.

Weakness:

I reckon the contribution of the paper as a density estimation method is marginal. Compared to the previously proposed transformers, [1,2], The key architectural difference is the use of a mixture of Gaussian as an output distribution instead of a discrete distribution generated from a softmax function. This change of output parametrization seems trivial and straightforward, compared to architectural improvements presented in [1, 2]. Another difference is the use of maximum mean discrepancy (MMD) as a regularizer (or an auxiliary objective function). However, as shown in Table 3, the gain from the use of MMD is not consistent.

The analyses conducted to evaluate generative models (contribution no. 2 on page 1) are valid, but the paper is not the first to perform such experiments and therefore it is not adequate to claim those experiments as a core contribution. Examining the predictive performance of a model trained on generated samples is used to evaluated generative adversarial networks [3, 4]. Also, measuring out-of-distribution detection performance is used widely in generative modeling literature [5, 6].

Remark 4 on page 6 is wrong. The test likelihood is an estimator of cross entropy between the model distribution and the data distribution and therefore is connected to Kullback-Leibler (KL) divergence between them. As long as KL divergence is a meaningful measure of discrepancy, test likelihood gives meaningful information. Also, the maximizer of the objective in (1) is indeed p(x), given infinite data and a correctly specified model.

There is no explanation of how the outlier classes in Pendigits, ForestCover, and Satimage-2 datasets are defined.

There is no description of what transformers are used as the baseline in Table 3. Also, what is the "standard Transformer" used in Table 4? How exactly do these transformers differ from TraDE?

Minor comments:

- It would be more appropriate to use Proposition instead of Lemma for Lemma 1 and Lemma 3.
- Currently, the numberings of lemmas and remarks are confusing. I suggest to number them separately.
- It would be nice to mention that the datasets used in the experiments are tabular data, just in case if a reader is not familiar to the datasets.

[1] Katharopoulos, Angelos, et al. "Transformers are rnns: Fast autoregressive transformers with linear attention." arXiv preprint arXiv:2006.16236 (2020).
[2] Child, Rewon, et al. "Generating long sequences with sparse transformers." arXiv preprint arXiv:1904.10509 (2019).
[3] Ye, Yuancheng, et al. "GAN Quality Index (GQI) By GAN-induced Classifier." (2018).
[4] Borji, Ali. "Pros and cons of gan evaluation measures." Computer Vision and Image Understanding 179 (2019): 41-65.
[5] Du, Yilun, and Igor Mordatch. "Implicit generation and modeling with energy based models." Advances in Neural Information Processing Systems. 2019.
[6] Grathwohl, Will, et al. "Your classifier is secretly an energy based model and you should treat it like one." arXiv preprint arXiv:1912.03263 (2019).

---

> ### Author Response · Authors · 2020-11-17
> **response to Reviewer 2**
>
> Thank you for your feedback. Please see the main comment above that addresses common concerns.
>
> We think the reviewer has understood our approach, appreciates its novelty and importance. The description of strengths of the paper is very positive so we are very surprised at the harsh score. If a paper is the first to apply a certain method, has a persuasive model, provides a rich view of how the model is effective, surely it deserves a score of more than 3. Please see the following for our response to specific comments.
>
> > I reckon the contribution of the paper as a density estimation method is marginal.
>
> We disagree with this strongly. It is undeniable that density estimation is an important problem. It is undeniable that Transformers have not been used for this problem previously. What is also undeniable is that our method works well for this problem. If a method works better on benchmark problems than almost a dozen existing papers, is it not in the best interests of the literature to have this result? The contribution of this paper is major and important: simple results that force us to re-evaluate how problems are solved are as important, if not more, than complex ones. This is especially pertinent for the modern practice of deep learning where there is an increasing tendency to make models more and more complex without rigorous investigation of the need to do so.
>
> > Compared to the previously proposed transformers, [1,2], The key architectural difference is the use of a mixture of Gaussian as an output distribution instead of a discrete distribution generated from a softmax function.
>
> Our work has little relation with [1,2]. [2] focuses on building a Transformer to work with very long sequences and [1] proposes a linear Transformer that is scalable and memory efficient in terms of the sequence length. These architectural innovations bear no relation to density estimation. The reviewer should not draw arbitrary connections to other publications using Transformers (this is a popular architecture). Just like a CNN architecture can be used for something other than classification, e.g. object detection, a Transformer can be used for something other than NLP tasks, e.g., density estimation. Just like the former is a novel/important contribution to the literature if it was not known previously, the latter is too.
>
> > The analyses conducted to evaluate generative models (contribution no. 2 on page 1) are valid, but the paper is not the first to perform such experiments and therefore it is not adequate to claim those experiments as a core contribution.
>
> We do not claim to invent these methods for evaluating generative models. These techniques are indeed known previously. And yet, none of the existing papers on density estimation use them. Our contribution is not to discover these techniques but to advocate their usage for density estimation. Since the reviewer clearly believes that this evaluation methodology is necessary, our paper that convincingly and rigorously makes the same point merits publication.
>
> > Remark 4 on page 6 is wrong. The test likelihood is an estimator of cross entropy between the model distribution and the data distribution and therefore is connected to Kullback-Leibler (KL) divergence between them. As long as KL divergence is a meaningful measure of discrepancy, test likelihood gives meaningful information.
>
> The reviewer is missing the point. Yes, the log-likelihood of a generative model is related to the KL-divergence. Remark 4 argues that the KL-divergence is simply a surrogate loss used to train a generative model. The value of KL is not directly related to performance on an actual downstream task for a generative model, e.g., outlier detection. Therefore KL-divergence alone is a not meaningful measure of discrepancy and alternative methods to evaluate generative models are necessary. The current literature on density estimation using deep networks does not use these alternate evaluation methods.
>
> > There is no explanation of how the outlier classes in Pendigits, ForestCover, and Satimage-2 datasets are defined.
>
> These datasets were already created to have inliers/outliers, as discussed in Shebuti Rayana ODDS library, 2016. http://odds.cs.stonybrook.edu which we have cited. The exact notion of outliers is different for different datasets; see http://odds.cs.stonybrook.edu/about-odds .This experimental setup is the same as that of Oliva et al, 2018.
>
> > There is no description of what transformers are used as the baseline in Table 3. Also, what is the "standard Transformer" used in Table 4? How exactly do these transformers differ from TraDE?
>
> We use the simplest Transformer model, namely the Encoder of Vaswani et al. 2017 in our paper; there is no decoder in our architecture. The exact setup of Table 3 is described in the narrative in Section 4.2.

---

### Official Review · AnonReviewer4 · 2020-10-27
**Density estimator with transformer architecture**

**Rating:** 4
**Confidence:** 4

**Review:**

# Summary
This paper uses the Transformer architecture for density estimation. It performs well on several non-trivial synthetic datasets and standard benchmark datasets. The authors also tested the model on other tasks that rely on density estimation.

## Pros
1. Addresses the issue of variable ordering on learning auto-regressive models and long-range dependencies with a new model architecture
1. Attempts to develop new evaluations other than log-likelihoods on practical uses of learnt density models
1. Impressive empirical results
1. Excellent written quality and comprehensive review of related literature.

## Cons
1. There is not much novelty. Simply borrowing the Transformer architecture does not seem sufficient for an academic venue
1. The additional evaluation tasks are not new, and the particular instances implemented have issues to be addressed
1. No mentioning of computational costs for training compared with benchmark methods.
1. Signs of heavy hyper-parameter tuning.


# Recommendation
Reject due mainly to a lack of novelty and some inadequate evaluations. I may raise the score slightly if the latter is addressed.

# Issues and questions:
1. Lemma 1: it's not clear why "variables with a cut edge" means. Do you mean "variables the removal of which disconnects a graph"?
1. How can the model take the entire sequence but only return conditional distributions $p(x_i|x_{1:i-1})$? I think I may be missing something here.
1. The equation of $alpha_j=\dots$ on page 4 is missing a parenthesis ")"
1. If the model is modelling each conditional correctly, why would it not model the joint well? This is mentioned in a strange place right after Lemma 3 which says it can model any joint.
1. All synthetic densities have a colour shift. A colour bar would help.
1. Introducing the MMD component does not seem to improve the log-likelihood much, so seems it is not essential for log-likelihood results. Can the author simply mention that adding this improves the model on small datasets?

## Questions regarding additional three empirical performance measures:
1. Why not conduct regression on all datasets?
1. Two-sample testing is not novel, and the results can be reported at the same place as log-likelihoods. There are also a few statistical testing paradigms which are more rigorous than simply reporting a classification acc (e.g. Liu et al 20).
1. OOD: how is the threshold swept? A better quantity to report would be the area-under-the-curve (AUC).
1. Robustness to noise: adding noise should definitely affect test likelihood. Why do the authors believe that a better model should be more robust to changes in the training dataset?
1. An overarching question is: are all the benefit of the proposed method a result of introducing the MMD regulariser?

---

> ### Author Response · Authors · 2020-11-17
> **Response to Reviewer 4**
>
> Thank you for your feedback. Please see the main comment above that addresses common concerns.
>
> >There is not much novelty. Simply borrowing the Transformer architecture does not seem sufficient for an academic venue
>
> See the common response to all reviewers above.
>
> > The additional evaluation tasks are not new, and the particular instances implemented have issues to be addressed
>
> We do not claim to invent these methods for evaluating generative models. These techniques are indeed known previously. And yet, none of the existing papers on density estimation use them. Our contribution is not to discover these techniques but to advocate their usage for density estimation. Since the reviewer clearly believes that this evaluation methodology is necessary, our paper that convincingly and rigorously makes the same point merits publication.
>
> > No mention of computational costs for training compared with benchmark methods.
>
> We only compared with NSF's training time and didn’t see a significant difference in terms of training time. Since each of these methods used different frameworks and tools (even different versions of the same framework e.g. tensorflow, pytorch) for implementation, it is not fair to compare their training time without significant changes to their codes.
>
> > hyper-parameter tuning
>
> The dataset (Table S3) used in this paper are widely different in terms of feature dimensions and number of training samples. In order to account for these differences, we used slightly different hyper-parameters for some of them. It is worth noting that all baselines that are used for comparison in this paper have utilized different hyper-parameters for different datasets too (e.g.  refer to Neural Spline Flows (Table 5), Autoregressive Energy Machines (Table 4))
>
> > Lemma 1: it's not clear why "variables with a cut edge" means. Do you mean "variables the removal of which disconnects a graph"?
>
> Yes.
>
> > How can the model take the entire sequence but only return conditional distributions
>
> At each position in the sequence, our model outputs the parameters of each conditional distribution. This is similar to how Transformers/RNNs can be used for language modeling in an autoregressive fashion:
> http://jalammar.github.io/illustrated-gpt2/#part-1-got-and-language-modeling
>
> > If the model is modelling each conditional correctly, why would it not model the joint well? This is mentioned in a strange place right after Lemma 3 which says it can model any joint.
>
> We are not entirely sure what the reviewer means by this. Can you please clarify?
> Here is an answer based on our understanding. Given finite data and an unknown model family for the true underlying distribution, our autoregressive MLE training only encourages TraDE to match the conditional distributions as closely as possible to the data. While it is true that the joint distribution would also be perfectly modeled if all conditional distributions were perfectly modeled, different types of estimation errors in the conditional distributions (inevitable with finite data) may lead to very different estimation quality of the full joint distribution corresponding to these estimated conditionals. We’ll clarify that Lemma 3 only states that TraDE has sufficient model capacity to approximate any joint distribution (not that it will estimate the joint distribution accurately from limited data, or any of the conditionals). The MMD regularizer is intended to improve the quality of the joint distribution estimates from finite data.
>
> > Can the author simply mention that adding MMD improves the model on small datasets?
>
> This is mentioned in the second paragraph in Section 4.2. We will add a similar sentence in the Introduction.
>
> > All synthetic densities have a colour shift. A colour bar would help.
>
> The color intensities in Figures 1 and 2 are not comparable. One of them (the true image) shows the grayscale intensity while the other one is simply a heatmap of samples from the learned distribution.
>
> > Why not conduct regression on all datasets?
>
> We used the HEPMASS dataset for experiments in Section 4.3. Results on other datasets are expected to lead to the same conclusion. We would like to note that the empirical evaluations in this paper are quite thorough and while these evaluation methods can be run on all datasets, the current experiments elucidate the point.

---

> > ### Author Response · Authors · 2020-11-17
> > **Continue response to Reviewer 4**
> >
> > > Two-sample testing is not novel, and the results can be reported at the same place as log-likelihoods. There are also a few
> > statistical testing paradigms which are more rigorous than simply reporting a classification acc (e.g. Liu et al 20).
> >
> > If the reviewer can provide the full citation for Liu et al. 2020 (what is the name of the paper?) we can fully understand this comment. To be clear, we do not claim to have invented the two-sample test. We simply advocate the use of the two-sample test [12] to distinguish between data generated from the learned distribution and real data.
> >
> > [12] Gretton, A., K. Borgwardt, M. Rasch, B. Schoelkopf and A. Smola: A Kernel Method for the Two-Sample-Problem. Neural Information Processing Systems, 2006.
> >
> > > OOD: how is the threshold swept? A better quantity to report would be the area-under-the-curve (AUC).
> >
> > Table 5 reports the average precision, i.e., the weighted mean of the precision at each threshold. The threshold t does not need to be swept explicitly. Because the dataset has ground-truth labels of outliers vs. inliers, we can simply sort the log-likelihood on samples and compute the integral under the AUC curve. See e.g., https://github.com/lupalab/tan/blob/master/tan/utils/anomaly_detection.py.
> >
> > > Robustness to noise: adding noise should definitely affect test likelihood. Why do the authors believe that a better model should be more robust to changes in the training dataset?
> >
> > Real data may contain outliers and a density estimator trained on real data should be insensitive to such outliers in the training set, e.g., see Altun & Smola 2006 [13]  for regularized density estimation or the field of robust statistics in general.
> >
> >
> > [13] Unifying Divergence Minimization and Statistical Inference via Convex Duality. Yasemin Altun and Alex Smola, COLT, 2006.
> >
> > > An overarching question is: are all the benefit(s) of the proposed method a result of introducing the MMD regulariser?
> >
> > Like all regularizers MMD does not universally improve the log-likelihood for all datasets, but it does produce sizable improvement for the HEPMASS dataset and it helps produce higher fidelity samples that are more similar to those in the dataset. This is seen in the regression experiment (Table 4). We also show experiments in Appendix B that MMD helps density estimation with limited data; this is common in biological applications, where experiments are often too costly to be extensively replicated (Krishnaswamy et al., 2014; Chen et al., 2020). Imposing the MMD regularization is not that computationally expensive with an efficient MMD implementation (training times were not noticeably longer with the MMD penalty added).

---

### Official Review · AnonReviewer1 · 2020-10-30
**Transformer-based Density Estimator (TraDE)**

**Rating:** 5
**Confidence:** 4

**Review:**

**Summary**
This work proposes a new auto-regressive density estimator built using self-attention module from the popular Transformer network. TraDE can be seen as an extension of decoder-only Transformer network where an input embeddings are given by a simple RNN-based encoder. Like Transformer, TraDE leverages multiple layers of self-attention module to implicitly model long-range dependencies. This effectively eliminates the need for explicit vertex ordering and hence useful on data with no known canonical ordering. The proposed model is general and can be applied to both continuous as well as discrete data. Along with the MLE objective, TraDE is additionally regularised using MMD penalty which can be easily back-propagated using reparametrization / Gumbel softmax trick.

On standard benchmark dataset, TraDE produces better density estimates compared to recently established state of the art baselines. To further evaluate the qualitative performance of density estimators, it proposes suite of various tasks on which TraDE is shown to work well.

**Quality**
The paper is very well written and easy to follow. The experimental evaluation followed are standard (for density evaluation) and additional tasks depicts the usefulness of the sampled samples.

**Originality**
As summarized above, TraDE is a simple extension of decoder of Transformer. Minor modification like RNN-based inputs (inspired by Wang et. al) and MMD loss led to dramatic improvement for density estimation tasks. However, overall the work lacks novelty and I feel it is only simple integration of different building blocks that produces better density estimator.

Moreover, the tasks used for qualitative evaluation are also not completely novel. As pointed by the author, some of them (regression, two sample test, OOD) are already employed by the prior work.

**Significance**
Despite lack of novelty this work demonstrates,
1. better empirical results as well as qualitative evaluation using various real world tasks, and
2. carefully integrated self-attention module can perform better than many complex density estimators.

**Clarity**
1. How do one sample data using TraDE. I do not find any mention of methodology employed for sampling.
2. Please include citations in Table 1.
3. From Table 3, I note that inclusion of MMD loss has very minor effect on performance. What is the training time tradeoff for including MMD loss ? Also compare quantitative results of (TraDE - MMD) model for various tasks.

---

> ### Author Response · Authors · 2020-11-17
> **Response to Reviewer 1**
>
> Thank you for your feedback. Please see the main comment above that addresses common concerns.
>
> >TraDE can be seen as an extension of decoder-only Transformer network.
>
> See the common response to all reviewers above.
>
>
> > work lacks novelty and I feel it is only simple integration of different building blocks that produces better density
>
> See the common response to all reviewers above.
>
>
> > How do one sample data using TraDE.  I do not find any mention of methodology employed for sampling
>
> As TraDE is an autoregressive estimator, we simply sample as is traditionally done with these types of models. To sample vector [x_1, x_2, x_3,...], we first draw x_1 ~ p(X_1), then x_2 ~ p(X_2|X_1=x1), then x_3 ~ p(X_3|X_2=x_2, X_1=x_1), etc. Here each of these conditional distributions is simply a mixture of Gaussians (for continuous data) or a categorical distribution (for discrete data).
>
>
> > MMD loss has a very minor effect, training time tradeoff for including MMD loss
>
>  Like all regularizers MMD does not universally improve the log-likelihood for all datasets, but it does produce sizable improvement for the HEPMASS dataset and it helps produce higher fidelity samples that are more similar to those in the dataset. This is seen in the regression experiment (Table 4). We also show experiments in Appendix B that MMD helps density estimation with limited data; this is common in biological applications, where experiments are often too costly to be extensively replicated (Krishnaswamy et al., 2014; Chen et al., 2020). Imposing the MMD regularization is not that computationally expensive with an efficient MMD implementation (training times were not noticeably longer with the MMD penalty added).

---

### Author Response · Authors · 2020-11-17
**Response to all reviewers**

We thank the reviewers for their feedback. The reviewers agree that TraDE demonstrates state-of-the-art empirical results that are better than a large number of competing approaches on several standard benchmarks across thorough experiments. We are glad that the reviewers resonate with our desire to develop a systematic evaluation framework for density estimation using downstream tasks.

**About our contributions and novelty.**

The simplicity here (what the reviewers consider “lack of novelty”) is a major asset of this paper. It is crucial to inform the density estimation community that such a simple technique is far more effective than many of the sophisticated state-of-the-art methods (e.g. flows) they have been researching. It is a reality check to help guide future research directions.

Similar papers on simple-but-effective discoveries have had high-impact and served as important eye-openers for ML subcommunities spanning pose estimation [3], natural language processing [4], zero-shot learning [5,6]. Note the reviewers’ main criticisms (“lack of novelty”, “merely simple adaptation of Transformer”) would also apply to the BERT paper [7], which revolutionized NLP, as it “merely” proposed a simple adaptation of the Transformer with two pretext tasks already known at that time [8,9]. Rather than introducing sophisticated models, the main contribution of that paper -and ours- is the discovery that these simple Transformer adaptations are remarkably effective in practice. Numerous other high-impact papers have also “simply” adapted Transformers in effective ways for other tasks [10,11].

TraDE is a simple and effective density estimation algorithm that works for both continuous and discrete valued data; it obtains significantly improved performance on standard benchmarks without requiring sophisticated architectural modifications. This is in contrast to normalizing flows (Durkan et al. (2019b;a); Kingma et al. (2016); De Cao et al. (2019), Nash & Durkan (2019), Papamakarios et al., (2017) ) that come with restrictive constraints on the input-output map such as invertible functions, Jacobian computational costs, etc. As compared to other auto-regressive models, TraDE can handle long-range dependencies and does not need to permute input features during training/inference like other methods such as Germain et al. (2015); Uria et al. (2014). Moreover, the objective/architecture of TraDE is general enough to handle both continuous and discrete data, unlike many existing density/distribution estimators. Finally, even though Transformer-like architectures have widely and primarily been used with discrete-valued data, to our knowledge, this is the first effective adaptation of it for continuous-valued density estimation.

We emphasize again that current state-of-the-art density estimation architectures are overly complex and have many limitations. Our key discovery in this paper is that a simple adaptation  of Transformers with our proposed changes can produce substantially better density estimates and therefore feel our findings are valuable to the community. To our knowledge, the effectiveness of Transformers/self-attention for density estimation has remained unknown until now.

[3] A simple yet effective baseline for 3d human pose estimation. Julieta Martinez, Rayat Hossain, Javier Romero, James J. Little; ICCV 17.

[4] A Simple but Tough-to-Beat Baseline for Sentence Embeddings. Sanjeev Arora, Yingyu Liang, Tengyu Ma; ICLR 2017.

[5] An embarrassingly simple approach to zero-shot learning. Bernardino Romera-Paredes, Philip Torr; ICML 2015.

[6] A Closer Look at Few-shot Classification. Wei-Yu Chen et al.; ICLR 2019.

[7] BERT: Pre-training of Deep Bidirectional Transformers for Language Understanding. Jacob Devlin, Ming-Wei Chang, Kenton Lee,
Kristina Toutanova; ACL 2019.

[8] Context2vec: Learning generic context embedding with bidirectional LSTM. Oren Melamud, Jacob Goldberger, Ido Dagan; CoNLL 2016.

[9] An efficient framework for learning sentence representations. Lajanugen Logeswaran and Honglak Lee; ICLR 2018.

[10] Image Transformers. Niki Parmar et al.; ICML 2018.

[11] Language Models are Few-Shot Learners. Tom Brown et al. arXiv 2020.

---

### Decision · Program_Chairs · 2021-01-07
**Final Decision**

**Decision:**

Reject

**Comment:**

This work explores an auto-regressive density estimator based on transformer networks. The model is trained via MLE with an additional MMD regularization term.
Various experiments are performed on small benchmarks and show good results on density estimation. It is great to see that such a simple model is indeed very effective for density estimation on various small benchmarks (such as 2D density estimation and MNIST).

The ablation experiments are informative and justify some of the model choices (such as the use of RNN to encode "positions"). Experiments are nicely chosen and paint a broad picture of the behaviour of the studied model.

The paper and author responses, however, excessively exaggerate the extent to which these results are relevant to the bigger picture in comparison to existing literature (e.g. flows and existing auto-regressive models).

As it has been extensively discussed with the reviewers, the proposed model is a straightforward application of a transformer network to auto-regressive modelling, this is specially so in light of existing work on auto-regressive models with transformers [e.g 1, 6, 8], self-attention [e.g 2]. BERT [7] itself can be used for auto-regressive modelling almost out-of-the-box (with the appropriate choice of masks during training).

At various points in the paper and author responses, it refers to flow models as "complicated/expensive" counterparts. These arguments are unfounded: auto-regressive models are particular cases of flows [3], and there are no obstructions to using transformer networks inside flows (in fact they have been already used, to achieve permutation equivariance and long-range correlations [e.g. 4]).
The paper leaves comparisons to spline-flows out, arguing they are "hard to implement". This is quite conspicuous, as not only spline-flows are straightforward to implement, they produce results entirely on-par with the presented model (as an example, look at Fig 2 from [9] in comparison to Fig 1 from this paper).
Finally, the paper also misses an important discussion about the computational complexity of the proposed method. Auto-regressive models are considerably slower to sample from in relation to other types of directed models. Even more so with transformer networks as conditioners. For instance, flows [3, 5] allow for substantially faster sampling of large-dimensional data relative to auto-regressive models (by exploiting parallel sampling).


Extra comments:

The paper says "... Self-attention also enables
permutation equivariance and naturally enables TraDE to be agnostic to the ordering of the features ... "
This is true only for a *single* conditional $p(x_i | \text{Transformer}(x_{0 \ldots (i-1)}))$, not for the *joint* density. It is actually not straightforward to build auto-regressive models that are permutation invariant or that incorporate other forms of domain knowledge in general.
As an example, see [4] for how transformers and spline-flows can be used to produce exact permutation-invariant densities.


[1] Chen, M., Radford, A., Child, R., Wu, J., Jun, H., Luan, D. and Sutskever, I., 2020, November. Generative pretraining from pixels. In International Conference on Machine Learning (pp. 1691-1703). PMLR.

[2] Parmar, N., Vaswani, A., Uszkoreit, J., Kaiser, Ł., Shazeer, N., Ku, A. and Tran, D., 2018. Image transformer. arXiv preprint arXiv:1802.05751.

[3] Papamakarios, G., Nalisnick, E., Rezende, D.J., Mohamed, S. and Lakshminarayanan, B., 2019. Normalizing flows for probabilistic modeling and inference. arXiv preprint arXiv:1912.02762.

[4] Wirnsberger, P., Ballard, A.J., Papamakarios, G., Abercrombie, S., Racanière, S., Pritzel, A., Rezende, D.J. and Blundell, C., 2020. Targeted free energy estimation via learned mappings. arXiv preprint arXiv:2002.04913.

[5] Huang, C.W., Krueger, D., Lacoste, A. and Courville, A., 2018. Neural autoregressive flows. arXiv preprint arXiv:1804.00779.

[6] Sun, C., Myers, A., Vondrick, C., Murphy, K. and Schmid, C., 2019. Videobert: A joint model for video and language representation learning. In Proceedings of the IEEE International Conference on Computer Vision (pp. 7464-7473).

[7] BERT: Pre-training of Deep Bidirectional Transformers for Language Understanding. Jacob Devlin, Ming-Wei Chang, Kenton Lee, Kristina Toutanova; ACL 2019.

[8] Child, R., Gray, S., Radford, A. and Sutskever, I., 2019. Generating long sequences with sparse transformers. arXiv preprint arXiv:1904.10509.

[9] Durkan, C., Bekasov, A., Murray, I. and Papamakarios, G., 2019. Neural spline flows. In Advances in Neural Information Processing Systems (pp. 7511-7522).